# Neurotoxic Effects of Neonicotinoids on Mammals: What Is There beyond the Activation of Nicotinic Acetylcholine Receptors?—A Systematic Review

**DOI:** 10.3390/ijms22168413

**Published:** 2021-08-05

**Authors:** Carmen Costas-Ferreira, Lilian R. F. Faro

**Affiliations:** Departamento de Biología Funcional y Ciencias de la Salud, Facultad de Biología, Universidade de Vigo, 36310 Vigo, Spain; maica.cf@hotmail.com

**Keywords:** neonicotinoids, mechanism of action, cholinergic system, mammals, neurotoxicity, central nervous system

## Abstract

Neonicotinoids are a class of insecticides that exert their effect through a specific action on neuronal nicotinic acetylcholine receptors (nAChRs). The success of these insecticides is due to this mechanism of action, since they act as potent agonists of insect nAChRs, presenting low affinity for vertebrate nAChRs, which reduces potential toxic risk and increases safety for non-target species. However, although neonicotinoids are considered safe, their presence in the environment could increase the risk of exposure and toxicity. On the other hand, although neonicotinoids have low affinity for mammalian nAChRs, the large quantity, variety, and ubiquity of these receptors, combined with its diversity of functions, raises the question of what effects these insecticides can produce in non-target species. In the present systematic review, we investigate the available evidence on the biochemical and behavioral effects of neonicotinoids on the mammalian nervous system. In general, exposure to neonicotinoids at an early age alters the correct neuronal development, with decreases in neurogenesis and alterations in migration, and induces neuroinflammation. In adulthood, neonicotinoids induce neurobehavioral toxicity, these effects being associated with their modulating action on nAChRs, with consequent neurochemical alterations. These alterations include decreased expression of nAChRs, modifications in acetylcholinesterase activity, and significant changes in the function of the nigrostriatal dopaminergic system. All these effects can lead to the activation of a series of intracellular signaling pathways that generate oxidative stress, neuroinflammation and, finally, neuronal death. Neonicotinoid-induced changes in nAChR function could be responsible for most of the effects observed in the different studies.

## 1. Introduction

Neonicotinoids (NNs) are an emerging class of synthetic pesticides that represent the largest market share worldwide [1]. There are at least seven commercial varieties of NNs, imidacloprid (IMI), thiamethoxam (TMX), clothianidin (CLO), thiacloprid (THI), acetamiprid (ACE), dinotefuran (DIN) and nitenpyran (NIT) [2,3]. This group of insecticides is increasingly used as substitutes for organophosphates and methylcarbamates, whose efficacy has declined considerably in recent years due to organism resistance and greater restrictions on their use due to toxicological considerations [4,5]. Currently, NNs are marketed in more than 120 countries and are among the most effective insecticides worldwide in the control of sucking insects and certain chewing species, in both agriculture and pets [1,2,6].

NNs were designed to be structurally similar to nicotine and, like this substance, they also act on the cholinergic neurotransmitter system [7]. Figure 1 presents the chemical structures of the main NNs in comparison with nicotine. Specifically, NN pesticides exert their neurotoxicity in target insects by acting as potent agonists of nicotinic acetylcholine receptors (nAChRs), which are widely distributed in the central nervous system (CNS) of these invertebrates [5,8]. The nAChRs are ligand gated ion channel that, when activated, allow small cations to pass through for several milliseconds before closing back to a resting or desensitized state [9]. NNs bind tightly to the alpha subunits of nAChRs and block them, causing neuromuscular paralysis and ultimately death [3,10,11].

The low affinity of NNs for mammalian nAChRs and a presumed low penetration through the blood–brain barrier (BBB) has led to the consideration that this class of insecticides has a relatively low toxicity for this group of vertebrates, compared to older insecticides such as organophosphates or carbamates [12,13,14]. This has led to an increasingly frequent replacement of traditionally used insecticides by NNs, whose use is quite widespread [1].

However, although NNs were previously thought to have a short biological half-life, recent studies have shown that these insecticides can remain in the environment for periods ranging from 1 day to almost 19 years [15]. Thus, humans are continuously exposed to these pesticides, which can be absorbed through food or drinking water, by inhalation or dermal contact with occupational spray by-products, as well as by indoor air contamination following termite or pet pest control treatments [4,16]. In line with this, residues of NNs have been detected in a wide variety of foods and surface waters, which can pose a serious health problem for the animals and humans that consume them [16,17,18,19,20,21,22,23,24,25,26].

In recent decades, various adverse effects of NNs on the health of non-target organisms have been reported, which has led to reconsideration of the supposed low toxicity of this class of insecticides. In particular, toxic effects of NN insecticides on insects of economic importance, such as bees or bumblebees, have been documented [27,28,29]. It is for this reason that in 2018 the European Union banned the use of several NNs in outdoor activities for posing a high risk to bees [30].

Although it has been widely assumed that NNs act selectively on insect nAChRs and have virtually no affinity for vertebrate receptors, data published in the literature have seriously challenged this assumption. Thus, in vivo and in vitro exposure of various animal species to this class of pesticides has been associated with mutagenic, neurotoxic and immunotoxic effects, as well as reproductive toxicity and endocrine system disruption [31,32,33,34,35,36,37,38,39,40].

Likewise, numerous studies have corroborated the chronic exposure of humans to this class of pesticides by detecting the presence of NNs or their metabolites in urine samples from children and adults [41,42,43]. Epidemiological studies revealed that the detection rates and quantities of these pesticides detected in human biological samples have gradually increased over the last decade, in proportion to domestic use [44].

This continued exposure is particularly alarming since it has recently been shown that both NNs and some of their metabolites can also act as agonists of human nAChRs, which are widely distributed in the body [45]. Consequently, exposure to this class of pesticides has been associated with the appearance of a wide variety of clinical symptoms in humans, including headache, fatigue, convulsions, abdominal and chest pain, weakness, tremors, respiratory failure or memory loss, among others [46,47,48,49,50,51]. In some cases, exposure to large amounts of NNs has produced very serious damage, leading to the death of the intoxicated person [52,53,54].

Taken together, the above information shows that the increasing use of NN insecticides can be a major problem, not only at the environmental level, but also for human health. Although NNs present low toxicity to non-target species, their presence in high quantities in all environments (soil, water, food) significantly increases the risk of exposure and toxicity to these species. The specific action of NNs on nAChRs seems to be responsible for most of the harmful effects that these substances can produce in humans and other mammals, although other possible mechanisms by which these pesticides produce their toxic effects on the nervous system have been investigated in recent years. Considering this information, in the present systematic review we investigate the available evidence of the last 10 years on the potential effects of NN pesticides on the nervous system of mammals. For this purpose, we analyze the main results obtained on the risks that exposure to this group of insecticides may cause to the structure and proper functioning of the nervous system of these non-target species.

## 2. Methodology

The present review was carried out with the objective of analyzing the results obtained in the most recent studies on the effects of NN pesticides on the mammalian nervous system. For this purpose, a systematic review was performed according to the guidelines established by Preferred Reporting Items for Systematic Reviews and Meta-Analyzes (PRISMA) [55]. The searches were carried out in the specialized databases PubMed, Scopus and Web of Science in May 2021, with a time restriction limited to studies published in the last ten years.

To identify the scientific articles that would be included in this review, the following search strategies were introduced in each of the databases: “((neonicotinoids) AND (nervous system) AND (mammals OR vertebrate OR rat))”; “((neonicotinoids) AND (mammals) AND (neurotoxic))”; “((neonicotinoids) AND (effect) AND (mammals))”.

### Exclusion and Inclusion Criteria

The articles included in this review met the following inclusion criteria: (1) original studies in article format; (2) in English or Spanish; (3) done in mammals; (4) studying the effects of the pesticide on the nervous system. Articles were excluded according to the following exclusion criteria: (1) theoretical articles or reviews and (2) studies in which NNs were administered in combination with other non-NN pesticides.

As a result of the searches carried out in the three databases, 678 articles published in the last ten years were identified. The articles were then exported to Refworks in order to eliminate those that were duplicated. In accordance with the inclusion and exclusion criteria, 466 titles and abstracts were screened to verify if they met the criteria previously mentioned. After this procedure, 423 articles were excluded for the reasons summarized in Figure 2. Finally, 38 articles were included in the present systematic review.

## 3. Results

The studies that have been analyzed in the present review show that in vivo or in vitro exposure to NN pesticides can induce the appearance of a wide range of neurotoxic effects in the different mammalian species studied. In general, most of the information gathered in these investigations demonstrates the capacity of NNs to alter the structure of the nervous system and, consequently, its correct functioning, both during development and in adulthood.

### 3.1. Effects of NNs on Rodents

Most of the research included here focuses on the study of the neurotoxic effects exerted by NN pesticides on the CNS of rodents. The results described are summarized in Table 1 and Table 2.

The toxic effects that NNs can produce in the brain depend on their ability to cross the BBB, which is the protection system that controls the exchange of endogenous and exogenous substances between the blood and the extracellular environment of the brain [56]. The studies analyzed confirmed that NNs do cross the BBB, based on the detection of pesticides and their metabolites in the brains of exposed animals [15,57,58,59,60].

So, in the study by Hirano et al. [58], after the administration of 5 mg/kg of CLO, variable concentrations of the pesticide and its metabolites were found in the brain. The levels detected in the brain reached values of 0.2 ppm in adult mice and almost 0.3 ppm in aged mice, approximately 5% of the total dose administered. In this research, the levels of CLO and its metabolites found in the brain were higher in the group of elderly mice compared to the adult group, while the concentrations of some CLO metabolites in the urine were lower in the elderly group. These results suggest that, because of the lower metabolizing capacity during aging, there is a delay in the excretion of NNs in this age group, which could explain their higher concentration in brain tissue.

Similarly, in the research by Katić et al. [15] after treatment with IMI for 28 days with doses of 0.8 mg/kg and 2.25 mg/kg, approximate levels of 0.3 and 0.5 μg/g, respectively, were detected in the brain. Burke et al. [57] observed that the treatment of rats with 0.5 mg/kg of IMI during pregnancy and the early postnatal stage was associated with brain concentrations of pesticide of 1.18 pmol/g in mothers and 0.044 pmol/g in offspring. These data suggest that NNs could be transmitted from mothers to offspring through the placenta during pregnancy and/or breast milk during the postnatal stage.

On the other hand, in the study by Kapoor et al. [59] after a single administration of 20 mg/kg of IMI in rats, peak concentrations in the brain were detected 12 h after exposure. The levels detected were 0.5 μg/g for IMI and approximately 1.6 μg/g for its metabolite 6-chlorine nicotinic acid, which would be 2.5% and 8% of the total IMI administered to the animal, respectively. Likewise, the team of Shamsi et al. [60] administered doses of 10, 20 and 40 mg/kg of ACE for 28 days and detected concentrations in the hippocampus of 4.6, 5.9 and 7.3 μg/g, respectively. In this study, it was also reported that the level of ACE in hippocampal tissue had increased in a dose-dependent manner after 28 days of treatment, suggesting its ability to bioaccumulate.

These studies show that NNs pesticides have the ability to cross the BBB reaching levels in brain tissue that can vary from 0.8% to 5% of the total dose administered. Although these studies are carried out under certain experimental conditions: with high doses, short treatment periods and limited to rodents, they are very useful for an estimation of the ability of NNs to cross the BBB of mammals. However, the mechanism by which these insecticides cross the BBB is unknown. In the study by Terayama et al. [61] the treatment with NNs did not alter the expression of cluster of differentiation 34 (CD34), which is a marker of vascular endothelial cells, one of the key cellular elements of the BBB. Previous studies have shown that CD34-positive cells can act to reduce BBB permeability [62], so it is possible that the increased permeability of the BBB to NNs is due to structural changes that do not necessarily involve endothelial cell involvement.

#### 3.1.1. Effects on Nervous System Development

Once in the brain, NN pesticides can exert different toxic effects, which vary depending on the time of life when exposure occurs. It has been shown that, if exposure occurs during development, NN pesticides can alter the transcription of numerous genes essential for proper neurodevelopment [63,64]. Consequently, incomplete development or brain hypoplasia may occur, characterized by reduced neurogenesis and altered morphology and distribution of neurons [64,65,66]. However, the team of Kimura-Kuroda et al. [64] found no alterations in the number or morphology of immature neurons and glial cells after treatment with ACE and IMI insecticides.

On the other hand, Kara et al. [11] documented an increase in the expression of the muscarinic acetylcholine M1 receptor after IMI treatment. The M1 receptor appears to play a key role in neuronal development and plasticity in the cerebral cortex, reaching a transient peak of immunoreactivity during early development [67]. Therefore, this alteration in M1 receptor expression may result in the appearance of alterations in the structure and synapses of neurons located in the cerebral cortex.

#### 3.1.2. Effects on Behavior and Cognitive Functions

NN pesticides produce various types of behavioral and motor effects in animals exposed chronically or acutely, during the embryonic period or in adulthood. These effects may occur due to alterations in different neurotransmitter systems, but mainly the cholinergic, dopaminergic and glutamatergic systems, which are systems that interact with each other to regulate neuronal activity and behavior.

It has been shown that exposure to CLO or IMI during the gestational and lactation period was associated with various alterations in behavioral parameters in several generations of offspring [57,68,69]. Behavioral alterations were also observed in adolescent and adult rodents treated with IMI and CLO, in particular, deficits in motor behavior were described, especially in animals exposed to the highest doses of the pesticides [58,70,71].

NN pesticides have a close similarity to nicotine, a cholinergic agonist whose effects on behavior are well studied and documented. For this reason, authors tend to compare the effects produced by NNs with those derived from nicotine exposure. Accordingly, it has been observed that the effects of NNs on behavior were very similar to those produced by nicotine, in which exposure to this alkaloid caused a decrease, but not a complete loss, of locomotor activity [72,73]. Given these similarities in the effects observed, the authors hypothesize that the changes in motor behavior induced by NNs are due, as occurs with nicotine, to an action of these pesticides on the nAChRs present in dopaminergic neurons. However, the team of Yoneda et al. [74] observed that exposure to DIN for 6 weeks increased locomotor activity in a dose-dependent manner. These differences could be because DIN might be less toxic to rodents than other NN pesticides such as IMI and, therefore, produce weaker effects in these animals [75,76].

Concerning mood, although adult mice rarely squeak in the human audible range, they may emit audible calls in stressful and crisis situations [77]. In the research of Hirano et al. [78], acute exposure of mice to a single dose of CLO (50 mg/kg) led them to emit human-audible vocalizations (4–16 kHz) as signs of excessive pesticide-induced stress or anxiety. Similarly, in this study the mice exhibited anxiety-like behavior. However, in the study by Yoneda et al. [74] chronic administration of various doses of the insecticide DIN did not produce significant changes in this behavior in mice.

As has been observed for locomotor behavior, the effects of NNs on anxiety were also compared with the effects produced by nicotine, which shows divergent results on anxiety levels in animals and humans. Thus, in some studies nicotine reduced both stress and anxiety [79,80,81,82], while in other investigations no effect was detected [83,84], and in other studies this compound even produced elevated levels of anxiety [85,86]. Like nicotine, NNs also seem to produce contradictory effects in the studies reviewed here. However, it is important to consider the differences in experimental design, doses used and treatment times, which differ between studies.

There is also discrepancy about the influence of NN pesticides on depressive behavior in rodents. While the team of Abd-Elhakim et al. [70] reported an increase in depression in animals after chronic treatment with IMI, Burke et al. [57] observed an antidepressant effect after early exposure to this pesticide. Similarly, Takada et al. [87] corroborated the antidepressant activity of the insecticide DIN after 5 weeks of exposure. These apparently contradictory results are like those observed in animals and humans exposed to nicotine in the short and long term [88,89,90] and are explained by considering the different sensitivities of nAChRs to agonists.

The cholinergic hypothesis of depression holds that depressive symptoms may arise as a result of cholinergic hyperactivity [91]. Therefore, the idea that, like nicotine, NN pesticides can produce antidepressant symptoms may be paradoxical, since they are cholinergic agonists. However, these observations can be reconciled by considering that these compounds, when chronically administered, can cause rapid and persistent desensitization of some nAChRs involved in mood regulation [92,93,94].

Specifically, nicotine is a desensitizing agent of α4β2 receptors and can produce a more potent and long-lasting desensitization of these receptors than acetylcholine itself [95]. Therefore, given the structural similarity with nicotine, it is to be expected that NNs are also capable of initially activating α4β2 receptors, but subsequently cause their desensitization and blockade when receptor stimulation is prolonged in time [96]. Thus, although it is probable that the antidepressant effect observed for some NNs is the result of the complex interaction of these compounds with different nAChRs, part of this effect could be due to desensitization and blockade of the α4β2. Consistent with this hypothesis, previous studies have shown that blockade of brain α4β2 receptors, especially those located in the amygdala, could decrease depression-like symptoms [97,98].

On the other hand, in the study by Burke et al. [57] it was shown that early exposure of mice to 0.5 mg/kg/day of IMI caused a decrease in aggression and an improvement in social dominance. This result is also similar to those found in studies on the effects of nicotine in mammals [99,100,101,102,103]. Similarly, modulation of the aggressive effect could be mediated by the action of nicotine or NN compounds on β2 receptors located in the prelimbic cortex [104].

Furthermore, chronic exposure to different doses of IMI or ACE altered learning and memory processes, especially in lactating rats [11,60]. This effect was similar to that observed with the exposure of animals and humans to nicotine [105,106]. These findings could be related to the essential role played by the cholinergic system in the hippocampus, the key brain structure in the processes of learning and memory formation [107].

In the hippocampus, the most abundant nicotinic receptor subtypes are α7 and α4β2 receptors [108]. Stimulation of α7 receptors triggers increased Ca^+2^ entry and activation of Ca^+2^-dependent signaling pathways that stimulate the release of glutamate. In this region, glutamate modulates synaptic plasticity and long-term potentiation, essential for learning and memory [109,110]. In line with this, it has been shown that exposure to NN pesticides decreases the expression of α7 receptors in the hippocampus [61]. Consequently, this is expected to severely affect the functioning of the hippocampal glutamatergic system, inducing changes, such as those observed in the study by Shamsi et al. [60], where treatment with ACE significantly reduced the levels of glutamate and its *N*-mehtyl-d-aspartate (NMDA)-like receptor subunits, which could translate into significant memory deficits.

It has also been shown that exposure to NN pesticides decreased neurogenesis in the dentate gyrus of the hippocampus and promoted apoptosis in this brain structure [66,111,112]. The dentate gyrus is one of the few brain areas where neurogenesis continues to occur during adulthood [113]. In addition, this region plays a fundamental role in various higher cognitive functions, such as learning and memory [114]. Therefore, the impairment caused by NN pesticides on the structure and function of the dentate gyrus could also cause the appearance of important learning and memory alterations.

#### 3.1.3. Changes in Neurotransmission

Neurotransmission is another process that is altered by exposure to NN pesticides. Since these pesticides exert their neurotoxicity by acting on nAChRs, it is to be expected that the cholinergic neurotransmitter system is the most affected in mammals. This system plays a key role in the regulation of cognitive functions such as learning, memory and attention, and its dysregulation is associated with the development of various neurodegenerative diseases [115,116].

In line with the above, Kapoor et al. [59] and Vohra et al. [117] reported that both acute and chronic administration of IMI caused a significant inhibition in acetylcholinesterase (AChE) activity in the rodent brain, although no significant effects have been observed in other studies [15]. Because of this inhibition, acetylcholine is not degraded and accumulates in the synaptic cleft, where it remains for a longer period. Consequently, the accumulated neurotransmitter can stimulate the nAChRs to a greater extent, causing their opening and the influx of Na^+^, K^+^ and Ca^+2^ ions across the membrane. Furthermore, as has been extensively demonstrated, the NNs can act as partial agonists of mammalian nAChRs, and thus may even compete with acetylcholine for binding to these receptors [118,119].

Therefore, an overstimulation of nAChRs by acetylcholine and/or NNs could cause a large influx of Ca^+2^ into the intracellular medium, such as that documented by Kimura-Kuroda et al. [120] in cerebellar neurons treated with ACE and IMI. Ca^+2^ is a universal intracellular messenger involved in the regulation of a wide range of activities, so that its intracellular levels are greatly reduced and tightly controlled [121]. Thus, any change in its cytosol levels can affect several processes, including neurotransmission, synaptic plasticity or gene transcription, among others [122,123].

Nitric oxide (NO) is an important physiological messenger synthesized by the enzyme nitric oxide synthase (NOS). There are at least three isoforms of this enzyme in the brain: the inducible NOS (iNOS), the endothelial NOS (eNOS), and the neuronal NOS (nNOS) [124,125]. Both eNOS and nNOS exhibit a mode of activation dependent on calmodulin binding to intracellular Ca^+2^ [126,127]. Therefore, an increase in cytoplasmic levels of this ion induced by NNs can lead to activation of both NOS isoforms and subsequent cytotoxic release of NO, as observed in the study by Duzguner and Erdogan [128].

On the other hand, there is a generally accepted paradigm that states that overstimulation of membrane receptors leads to a reduction in their number in the cell membrane [129]. Therefore, this paradigm would explain how overstimulation of brain nAChRs by the agonist action of acetylcholine or NNs, can cause a decrease in the number of nAChRs in different brain regions, as demonstrated in the research of Terayama et al. [61]. However, the opposite results were obtained in animal and human studies chronically exposed to nicotine, as an increase in nAChR density (commonly known as upregulation) was observed [130]. Therefore, further research is needed to understand the effects of chronic exposure to NNs on both the number and functional status of nAChR receptors.

The interaction of NN insecticides with cholinergic receptors was also associated with alterations in the dopaminergic system. Dopaminergic neurons are involved in a wide range of functions, such as voluntary movement, motivated behavior, working memory or affectivity [131,132,133]. It has been shown that acute administration of CLO or IMI caused a marked increase in dopamine levels in the rat striatum [134,135,136]. In the study by Faro et al. [134] it was found that CLO-induced dopamine release was mainly mediated by an exocytotic-, vesicular-, and Ca^+2^-dependent mechanism. In their subsequent investigation, Faro et al. [135] concretized their results, demonstrating that CLO-induced striatal dopamine release requires activation of nAChRs possessing the α4 or α7 subunits, but not the β2 subunit. This could be related to the fact that NNs, like acetylcholine, bind to the α-subunits of nAChRs, whereas the other subunits act as regulators. Therefore, the administration of antagonists of α-subunits will prevent the interaction of the pesticide with the receptor binding site and, consequently, blocks its effects on in vivo dopamine release. Likewise, Faro et al. [135] demonstrated that the increase in dopamine release also appears to depend on the activation of muscarinic acetylcholine receptors (mAChRs). On the other hand, the team of Abd-Elhakim et al. [70] found opposite results, as chronic exposure to the IMI for 60 days reduced the total dopamine content in brains from adult and adolescent rats.

Changes in dopamine release induced by exposure to NN pesticides could furthermore be related to changes in the transcription of the enzyme tyrosine hydroxylase (TH). TH is the rate-limiting enzyme of catecholamine synthesis and responsible for catalyzing the hydroxylation of tyrosine to l-3,4-dihydroxyphenylalanine (l-DOPA) [137]. Yoneda et al. [74] and Kawahata and Yamakuni [136] have found that in vivo and in vitro administration of NNs, both short- and long-term, enhanced transcription and increased phosphorylation of brain TH, which was associated with an increase in its activity. In this regard, Kawahata and Yamakuni [136] found that the insecticide IMI facilitated TH transcription by acting as a partial agonist at α3β4 and α7 receptors. Because of its interaction with these receptors, it is considered that IMI could cause long-term activation of intracellular signaling of both cAMP response element-binding protein (CREB) and protein kinase A (PKA)/extracellular signal-regulated kinase (ERK) and Rho cascade. Activation of these signaling cascades could facilitate TH transcription and expression of the phenylethanolamine N-methyltransferase (PNMT) gene, the enzyme that catalyzes epinephrine biosynthesis, leading to up-regulated catecholamine production and secretion.

On the other hand, exposure to NNs also affected the functioning of the glutamatergic neurotransmitter system. Glutamate is the main excitatory neurotransmitter in the CNS and plays critical roles in learning, memory and cognition [138,139]. Both excessive glutamate release and glutamatergic dysfunction are associated with multiple neurodegenerative diseases and psychiatric disorders [140,141]. In the study by Shamsi et al. [60] it was observed that exposure to the pesticide ACE for 28 days significantly reduced glutamate levels in the hippocampus of the treated subjects. In addition, treatment with the highest doses of ACE (20 or 40 mg/kg) caused a decrease in the expression of NR1, NR2A and NR2B genes. These genes encode the subunits that make up the NMDA ionotropic glutamatergic receptors, which are ligand-activated cation channels that allow Ca^+2^ and Na^+^ to enter and K^+^ to exit the cell [139,142].

Other systems that suffered alterations after exposure to NNs were the serotonergic and GABAergic systems. On the one hand, the serotonergic system is one of the oldest neurotransmitter systems in the brain and plays a key role in mood regulation, as well as participating in other basic biological functions including appetite, sleep and cognitive function [143,144]. The GABAergic neurotransmitter system is fundamental for the neurogenesis processes in the developing CNS, while in the mature CNS it is involved in functions such as the regulation of neuronal responsiveness, the maintenance of the excitation/inhibition balance and neuronal plasticity [145,146]. Changes in any of these neurotransmitter systems has been related to various pathologies, such as schizophrenia due to deficient functioning of the GABAergic system, or depression in the case of abnormalities in the serotonergic system [147,148].

In line with the above, Abd-Elhakim et al. [70] have reported that exposure to IMI for 60 days significantly reduced GABA and serotonin levels in the brain of adolescent and adult rats. However, in the study by Takada et al. [87], treatment with DIN for 5 weeks was not associated with a decrease in the number of serotonergic neurons. These differences are possibly linked to differences between studies, as the experimental designs differed in the type, amount, exposure time, and animal species. Therefore, further studies are needed to clarify this controversy regarding the effects of NN compounds on serotonergic neurotransmission.

#### 3.1.4. Induction of Oxidative Stress

Oxidative stress is one of the mechanisms by which pesticides can exert their toxicity [149,150]. Oxidative stress occurs when the production of reactive oxygen species (ROS) or nitrogen species exceeds the capacity of the body’s antioxidant system to scavenge them [151]. As a consequence of the overproduction of free radicals, lipid peroxidation (LPO), DNA damage, and extensive protein oxidation occur, leading to cellular degeneration [151,152]. The brain, with its high lipid content, high oxygen consumption and reduced antioxidant capacity, is particularly susceptible to oxidative stress [153,154].

Exposure to NNs has been associated with an increase in ROS levels and, consequently, LPO, oxidative damage to genetic material and protein oxidation [70,128,155,156]. In particular, an increase was observed in the levels of malondialdehyde (MDA), 8-hydroxyguanosine and protein carbonyl, which were used as markers of LPO, DNA oxidation and protein oxidation, respectively [70,156].

To counteract the overproduction of ROS, the body is provided with a natural antioxidant system formed by a set of endogenous antioxidant enzymes and other non-enzyme compounds. In addition, this system is reinforced by exogenous antioxidants obtained through the diet [157,158]. Together, both endogenous and exogenous antioxidants aim to minimize the levels of ROS, while at the same time ensuring the presence of sufficient levels of these molecules to perform cell signaling and redox regulation functions [159,160].

It has been shown that exposure to NNs for approximately one month caused an increase in the activity of the enzymes catalase (CAT) and glutathione peroxidase (GSH-Px) in the brain [15,128]. However, a longer treatment with NNs decreased the antioxidant capacity, decreasing the activity of these two antioxidant enzymes [70,156]. These results suggest that, although in the short term these antioxidant defense systems attempt to counteract the intense generation of ROS induced by NNs, when this overproduction is maintained over time there is a depletion of these protective molecular systems. On the other hand, during long-term exposure to the pesticide ACE, there was an increase in superoxide dismutase (SOD) and glutathione-S-transferase (GST) activity in the mitochondrial matrix [156]. These data suggest that these enzymes maintain their activity over time in an attempt to reduce excessive ROS levels and remove the end products of LPO.

Glutathione (GSH) is another endogenous antioxidant that plays an important role in the detoxification of ROS [161]. GSH has a strong ability to interact with ROS and neutralize them, making it a potent antioxidant [162]. The studies included here have shown that chronic exposure to NNs for 1 or 6 months resulted in a significant decrease in GSH levels [128,156], suggesting that GSH depletion could take place already at the early stages of pesticide exposure and would no longer recover over time.

Xanthine oxidase is an enzyme that is widely distributed in various mammalian tissues, including the brain [163]. This enzyme is involved in the oxidation of a wide variety of endogenous and exogenous substrates, such as purines or ethanol. As a result of its activity, xanthine oxidase increases ROS levels by generating superoxide radicals and hydrogen peroxide [164,165,166]. In line with this, in the study by Duzguner and Erdogan [128], exposure to IMI caused an increase in xanthine oxidase activity, which, consequently, could contribute to increased cellular oxidative stress through the production of ROS.

The mitochondria are the main ATP producing center in cells, and during the process of synthesis of these molecules a certain amount of free radicals, considered physiological, are generated [167]. Under conditions of oxidative stress, mitochondria become one of the main targets of ROS damage, which can induce mutations in mitochondrial DNA, alter their antioxidant defense systems, damage the mitochondrial respiratory chain, affect membrane permeability or influence Ca^+2^ homeostasis [168]. Some of the studies analyzed here show that the exposure to NN pesticides could induce several of these alterations in the rodent CNS neurons.

It has been shown that treatment with NN pesticides increased mitochondrial membrane permeability [155,156]. These changes could be due to the mitochondrial membrane LPO induced by ROS. Another change observed was the reduction in the mitochondrial membrane potential [155]. Since mitochondria act as reservoirs of Ca^+2^ ions, excessive generation of ROS can trigger the release of these ions into the cytosol. The increase in Ca^+2^ levels also change the mitochondrial potential that leads to the production of superoxide radicals, thus initiating a vicious circle [168].

Thus, in the situation of oxidative stress caused by exposure to NN pesticides, the mitochondrial antioxidant defense systems are insufficient to counteract the oxidative damage. So, the excessive ROS can attack the lipids of the mitochondrial membranes, causing prolonged opening of the mitochondrial permeability transition pore and altering the mitochondrial membrane potential [169,170]. As a consequence, the structure of these organelles is severely damaged leading to mitochondrial dysfunction and a drop in the rate of oxygen consumption, as observed in the study by Gasmi et al. [156].

#### 3.1.5. Induction of Inflammation

The inflammatory process plays an essential role in helping the immune system to counteract pathological states such as infection, but when inflammation becomes unbalanced or prolonged over time it can lead to progressive tissue damage [171]. In the CNS, the inflammatory response is mediated by the release of cytokines, chemokines, secondary messengers and ROS [172]. Most of these mediators are synthesized and released by microglia and astrocytes in the CNS [173]. Thus, when an insult occurs due to the action of neurotoxic substances such as pesticides, the microglial cells are activated and no longer exhibit a branched morphology but rapidly manifest an amoeboid phenotype [174,175]. Likewise, the astrocytes also become reactive in response to insults and there is an upregulation of glial fibrillary acidic protein (GFAP), which is the main component of the intermediate filaments of the astrocytes [176].

Activation of microglia involves a series of functional and morphological changes in these cells. Two main polarization states or phenotypes have been described and that show opposite characteristics: the M1 cytotoxic-proinflammatory phenotype or the M2 cytoprotective-anti-inflammatory phenotype [177,178]. M1 or cytotoxic microglia produces ROS and proinflammatory cytokines, increasing tissue injury and creating an adverse environment for neuroprotection [179]. In contrast, activation to the M2 phenotype is related to the production of growth factors and anti-inflammatory cytokines that stimulate tissue regeneration [180].

Some of the studies reviewed here have shown that exposure to NN pesticides can trigger an extensive inflammatory process. Thus, in response to CNS insult, astrocytes became more reactive, evidenced by a marked increase in GFAP levels [70]. Similarly, microglia acquired an amoeboid morphology and increased expression of its marker Iba1, which is a hallmark of the activation of this cell type [65,66]. This glial activation was related to the increased release of the proinflammatory cytokines interleukin 1beta (IL-1β), interleukin 6 (IL-6), tumor necrosis factor alpha (TNF-α), and interferon gamma (IFN-γ), observed in the study by Duzguner and Erdogan [128]. However, in this same study there was a decrease in the levels of the cytokine interleukin 12 (IL-12), also a promoter of the inflammatory process. Taken together, these results suggest that exposure to NN pesticides promotes microglial polarization to the M1 cytotoxic phenotype, which could promote neurotoxicity by inducing excessive release of proinflammatory cytokines, as observed in the studies of Kagawa and Nagao [65] and Nakayama et al. [66].

Another process that takes place during neuroinflammation is the activation of iNOS, which is expressed in several cell types, including microglia, astrocytes and CNS neurons [181]. This enzyme is responsible for the production of NO, which plays an important role as a neurotransmitter and neuromodulator in the CNS [182]. The expression of iNOS in the CNS is tightly regulated, although various intrinsic and extrinsic stimuli can induce its activity [181]. In this regard, it has been shown that, during the inflammatory response, some cytokines such as IFN-γ can activate signaling pathways that stimulate iNOS activity and NO production [183,184]. However, excessive stimulation of iNOS can lead to massive production of NO and NO-derived peroxynitrate, which contribute greatly to oxidative stress [185]. The results of the study by Duzguner and Erdogan [128] demonstrated that exposure to IMI enhanced iNOS mRNA transcription and consequently led to a significant increase in NO production.

#### 3.1.6. Effects on Energy Metabolism

In addition to the alterations mentioned in the previous sections, exposure to NNs was also associated with alterations in the metabolism of brain cells. The brain is an organ with an unusually high metabolic demand, since it represents only 2% of the body mass, it uses approximately 20% of the total glucose and oxygen of the human organism [186]. Thus, strict regulation of cellular energy metabolism is essential to ensure normal brain function, and any alteration in this can be associated with the development of various pathologies [187,188]. Some in vitro and in vivo investigations have shown that both short- and long-term exposure to IMI and CLO pesticides altered the metabolism and energy balance of rodent brain cells, especially in the hippocampus [112,189]. These changes could reflect the metabolic plasticity of brain cells, that is, an attempt by brain cells to adapt their metabolism to the adverse conditions generated by NN pesticides. However, these significant alterations may potentially contribute to neuronal dysfunction and induce apoptosis [190].

#### 3.1.7. Induction of Apoptosis

Conditions of oxidative stress and neuroinflammation, mitochondrial damage, alterations in neurotransmitter systems, as well as changes in energy metabolism induced by NN pesticides can eventually lead to neuronal death. Some of the studies included here agree that both acute and chronic treatment with IMI and ACE pesticides caused DNA damage and neuronal degeneration, especially affecting the hippocampus [15,60,70,112,155].

Of the neurogenic regions in the adult CNS, it is the hippocampus that receives the most attention, since during hippocampal neurogenesis new granular cells are generated in the dentate gyrus, which are essential for the correct functioning of higher cognitive processes, such as memory or mood regulation [191,192]. Thus, a dysregulation in neurogenesis in the adult hippocampus has been linked to the development of various neurological diseases such as Alzheimer’s disease and mood disorders [193,194]. In line with this, Nakayama et al. [66] have observed that early exposure to ACE or IMI reduced neurogenesis in the dentate gyrus of the hippocampus in mice. Similarly Shamsi et al. [60] demonstrated that ACE exposure induced significant neuronal degeneration and apoptosis in this region.

Although there are different modes by which cell death can occur in the brain, apoptosis appears to be the main mode of cell death induced by NNs. Apoptosis can be triggered by two main pathways: the extrinsic (or death receptor) pathway and the intrinsic (or mitochondrial) pathway. The extrinsic pathway is triggered outside the cell by death ligands, which bind to and activate death receptors located on the cell membrane [195]. Activation of these receptors leads to activation of caspase-8, which cleaves other downstream caspases and ultimately causes cell death [196]. In line with this, it has been shown that exposure to NN pesticides enhanced the expression of TNF-α, which can act as a death ligand and trigger the extrinsic pathway of apoptosis [128].

The intrinsic or mitochondrial pathway of apoptosis is usually triggered inside the cell in response to noxious stimuli, such as excessive increases in intracellular ROS and Ca^+2^ concentrations [197]. These cellular stress factors trigger the activation of the Bax protein to form pores in the outer mitochondrial membrane and release cytochrome C, which triggers the activation of caspases in the cytosol and apoptosis [198]. As previously mentioned, treatment with NN compounds induced an increase in intracellular Ca^+2^ concentration and increased oxidative stress, two conditions closely related to the activation of the mitochondrial pathway of apoptosis [120,128,155,156].

Some of the research has shown that ACE pesticide exposure caused neuronal death through apoptosis [60,155]. Likewise, the team of Abd-Elhakim et al. [70] observed that exposure to IMI increased the levels of the pro-apoptotic protein Bax, while reducing the concentration of the anti-apoptotic protein Bcl-2, which under physiological conditions, keep apoptosis-inducing proteins, such as Bax inactive [199]. These results suggest that, possibly due to adverse conditions caused by NN pesticides in the brain, both the intrinsic and extrinsic pathways of cellular apoptosis may be activated.

### 3.2. Effect of NNs on Humans

Available studies on the effects of NN pesticides on humans are limited and only seven studies in total have been identified. The results are summarized in Table 3 and Table 4.

#### 3.2.1. In Vitro Neurotoxic Effects on Human Cell Cultures

A total of three in vitro studies have been identified evaluating the effects of exposure of the human neuroblastoma cell line SH-SY5Y to the pesticides IMI, ACE, CLO, THI and/or TMX over a 24- or 48-h period.

In the study by Hirano et al. [200] different concentrations of CLO (1–100 μM) were administered to SH-SY5Y cells expressing α3, α7, β2 and β4 nAChRs subunits. The pesticide caused a dose-dependent increase in the number of cells and the growth of their neurites. According to the authors, this effect was due to the agonist action of CLO on nAChRs, indicating that the pesticide was able to activate these receptors and increase Ca^+2^ entry into the cell interior. This increase in intracellular Ca^+2^ levels could be responsible for the ERK activation caused by the pesticide [201]. ERKs are important components of signaling cascades that regulate a variety of cellular activities in the CNS, including cell proliferation, migration and differentiation [202]. In line with this, some of the metabolites of NNs have been shown to have a higher affinity for human nAChRs and can induce higher ERK activation than the parent compound [203]. Thus, further studies investigating the potential effects of the metabolites of these NN pesticides should also be conducted.

Another finding documented in the study by Hirano et al. [200] was that CLO exposure downregulated several genes related to neuronal function and morphology. These changes could underlie the increase in neurite length that was observed during neuronal differentiation. In contrast, in the study by Cheng et al. [204] administration of higher concentrations (10–2000 mg/L) of IMI, ACE and TMX caused significant inhibition of cell growth and viability. Similarly, in the research of Şenyildiz et al. [205] the exposure to IMI, ACE, CLO, THI and TMX (0.05–4 mM) induced neuronal DNA damage, especially at the higher concentrations. Furthermore, the results of both studies show that THI and TMX are significantly more toxic than the other NNs used in the studies.

Taken together, the results of the in vitro studies suggest that, although excessive amounts of NNs can cause cell death, exposure to low concentrations could have proliferative effects. As has been observed for other mechanisms of neurotoxicity discussed in the present review, the results of the in vitro studies described here are consistent with those described for nicotine under similar experimental conditions [206,207].

#### 3.2.2. Descriptive and Analytical Studies

The neurotoxic effects that NN pesticides can produce in the nervous system depend on their ability to cross the BBB. The team of Vinod et al. [208] described the case of a 23-year-old man who deliberately ingested THI mixed with alcohol. The intoxication manifested with a wide range of symptoms, including status epilepticus and profound unconsciousness, suggesting the ability of NNs to cross the human BBB.

Similarly, in the study by Marfo et al. [209], a relationship was found between environmental exposure of the general population to ACE and TMX pesticides and the appearance of a symptom picture called “neonicotinic symptoms”. This set of symptoms was characterized by loss of recent memory, headache, generalized fatigue, tremor of the fingers or muscle weakness, among others. Some of these symptoms could reflect, again, the ability of NN molecules to penetrate the CNS by crossing the BBB. Other neurological symptoms such as disorientation, fever, dizziness or drowsiness were also reported in studies of people intoxicated with IMI [210,211,212].

On the other hand, some studies have suggested that exposure to NN pesticides during pregnancy may be related to an increased risk of developing autism spectrum disorders (ASD) and a lower IQ of the offspring, although these results should be interpreted with caution [213,214]. In this regard, several studies provided evidence of a possible association between neurodevelopmental disorders and exposure to environmental chemicals such as pesticides, especially during the gestational period [215,216]. For it has been shown that fetuses, infants and children are especially vulnerable to exposure to toxic chemicals in the environment [217,218].

### 3.3. Effects of NNs on Other Mammals

Only three studies have described the role of NNs on mammals other than rodent and humans. Specifically, one of the investigations focuses on the toxic effects of IMI on adult female white-tailed deer *(Odocoileus virginianus)* and their offspring, while the remaining two studies evaluate the impact of this pesticide on the great roundleaf bat (*Hipposiderosa armiger terasensis*). A summary of the results described in these articles is shown in Table 5.

In the study by Berheim et al. [219] the presence of the pesticide IMI was detected in the body of all deer, both in those treated with the pesticide and in the control group. These results indicate the presence of NNs in the environment, mainly in food and vegetation. In addition, although IMI was detected in many of the organs of white-tailed deer, the values detected in the brain were low or undocumented. These findings were attributed to the inability of the pesticide to cross the BBB in these animals.

Research by Hsiao et al. [111] and Wu et al. [220] evaluated the impact of chronic IMI treatment on the echolocation system of bats. Echolocation is an orientation system used by some animals by which they gather information from the environment by making use of echoes [221]. This system allows bats to orient themselves through complex environments and to locate and identify objects in complete darkness [222,223]. It has been shown that, after exposure to IMI insecticide, bats were disoriented and showed irregular flight paths [111,220].

In addition, exposure of bats to IMI was associated with a marked increase in neural apoptosis in the hippocampal CA1 area and the medial entorhinal cortex [111,220]. As previously commented, this apoptotic pathway is initiated by an increase in the permeability of the mitochondrial outer membrane and the release of various proteins, such as cytochrome C, into the cytoplasm [224]. Consequently, effector caspases, such as caspase 3, are activated by cleavage events and collectively orchestrate the execution of apoptosis [225]. In this regard, the team of Wu et al. [220] documented the enhanced expression of apoptotic cytochrome C and caspase 3 molecules in bat cerebral cortex tissue.

On the other hand, stress conditions that interfere with endoplasmic reticulum (ER) homeostasis can also initiate a process of apoptosis different from the one previously discussed [226]. During ER stress-mediated apoptosis, caspase 12 translocate from the ER to the cytosol, where it cleaves and activates pro-caspase 9, which, in turn, activates caspase 3 or effector caspase [227]. Thus, it has been shown that exposure to IMI increased the expression levels of caspase 12 and caspase 3 in bat cochlear tissue [220].

In addition, the team of Wu et al. [220] also detected an increase in TNF-α expression in both cortical and cochlear tissue of bats. TNF-α is an apoptosis-inducing death ligand, so the increase in its levels may be directly related to activation of the extrinsic pathway of apoptosis and, alternatively, may also induce activation of the intrinsic or mitochondrial pathway of apoptosis [228].

Taken together, these data suggest that one of the mechanisms by which IMI could alter the echolocation system of bats is through cell death in brain areas involved in spatial memory processing. Specifically, it appears that IMI-induced apoptosis may occur through mitochondrial dysfunction in the cerebral cortex and ER stress in the cochlea.

In the research of Wu et al. [220], IMI administration to bats also reduced the expression of brain proteins related to echolocation. On the one hand, otoferlin is a protein expressed by sensory hair cells that plays a crucial role in auditory function and is expressed in the brain and inner ear [229,230]. This protein plays an important role in the transmission of sound signals in the brain during echolocation [231]. It has been shown that IMI exposure decreased the expression of the hearing-related orthophelin in the cochlea and inferior colliculus [220].

Wu et al. [220] also found that the pesticide reduced the expression of prestin, a cochlear protein related to hearing. Prestin is a membrane motor protein localized to outer hair cells that is essential for cochlear amplification and is widely expressed in echolocating animals [232,233,234]. Similarly, exposure to IMI downregulated the expression of Forkhead box protein P2 (FOXP2), a voice-related protein in the superior colliculus [220]. FOXP2 is a transcription factor involved in the coordination of orofacial movements required for speech and whose alteration is associated with language disorders [235]. Thus, it is considered that FOXP2 may be involved in the neural circuits that support echolocation in bats [236,237].

Therefore, the set of results suggests that IMI treatment interferes with the orientation system and the spatial, auditory and vocal memory of bats, which are essential for the correct functioning of their echolocation ability.

### 3.4. Discussion

Most of the studies analyzed were conducted in rodents and provided valuable information on the biochemical mechanisms by which NN pesticides produce toxicity in the mammalian nervous system. First, sufficient evidence has been collected on the ability of these insecticides to cross the BBB and reach the CNS, as evidenced by the presence of NNs in the brain tissue of rodents or the neurological symptoms observed in humans. Once in the brain, the NNs produce important neurotoxic effects, which appear to vary depending on the vital moment in which the exposure occurs.

The results show that many of the neurotoxic effects observed after exposure to NN pesticides are directly or indirectly related to their action on nAChR receptors. Although NNs have low affinity for these receptors, the great variety and ubiquity of nAChRs, present from protozoa to higher eurokaryotes, adds to their diversity of functions, such as the regulation of neuronal development, the modulation of various neurotransmission systems, the learning, memory, and emotions, increases the risk of NNs toxicity in the nervous system.

The nAChRs are widely distributed throughout the mammalian CNS, but especially in areas such as the cortex, striatum, hippocampus, thalamus, hypothalamus, amygdala, ventral tegmental area, cerebellum or some nuclei of the brain stem [238,239]. Some of the studies included in the present review demonstrated that the most commonly expressed subtypes are the heteromeric α4β2 type and the homomeric α7 type receptors [240], being these receptors responsible for the effects of NNs in the mammalian nervous system. Figure 3 summarizes the main neurotoxic effects induced by the NNs described in the studies analyzed in this review.

Anatomical and functional evidence suggests that the α4β2 and α7 type receptors are both pre- and postsynaptic and their activation promotes the entry of Ca^+2^ into the neuronal interior. In this way, the activation of these receptors and the influx of Ca^+2^ into the neuron can modulate the release of almost all neurotransmitters, both excitatory and inhibitory [238]. This could explain, at least partially, the alterations in the different neurotransmission systems observed after exposure to NN pesticides.

It has been shown that, after a short exposure to nicotine, the α4β2 receptors becomes desensitized, whereas α7, being less sensitive, remains functional even when exposed to very high concentrations of this agonist [241]. The α7 receptors allow the Na^+^ entry and are highly permeable to Ca^+2^, while the α4β2 receptors, although they allow the Na^+^ entry through them, have a lower permeability to Ca^+2^ [242]. On the other hand, most studies evaluate the effects of NNs administered chronically in periods ranging from 2 to 24 weeks of exposure. Thus, prolonged exposure may induce changes in the two receptor subtypes in a similar way to nicotine, desensitizing and closing the α4β2 receptors, while the α7 receptors remain active.

Some of the studies analyzed in this review show that one of the main effects produced by NNs is the increase in the influx of Ca^+2^ in both rodent neurons and human cells in vitro. Ca^+2^ is a general intracellular messenger and controls virtually all aspects of neuronal function (e.g., metabolism, synaptic transmission and plasticity, gene transcription, programmed cell death) [121], and cells possess mechanisms responsible for maintaining its intracellular concentrations at sufficiently low levels [243]. Therefore, the increase in the Ca^+2^ levels in the cytosol can trigger multiple biochemical effects in the cell, as show in Figure 4.

Furthermore, the NNs-induced nAChRs activation, mainly of α7 subtype, in addition to increasing Ca^+2^ entry, also increases Na^+^ influx, which depolarizes the neuronal membrane and opens voltage-gated ion channels, including the voltage-dependent calcium channels (VDCC), increasing even more the intracellular Ca^+2^ concentrations. This Ca^+2^ of extracellular origin can bind to ryanodine receptors, which are Ca^+2^ channels present in the ER membrane, causing their opening and the release of Ca^+2^, further increasing its level in the cytosol [243]. On the other hand, an increase in the Ca^+2^ levels and its binding to calmodulin can activate the eNOS and nNOS, which contribute to oxidative stress, as observed in several of the studies analyzed. The excessive Ca^+2^ levels can also promote the activation of calcium-binding calpain proteases, which can lead to the degradation of proteins and structural enzymes [126,244].

Exposure to NN pesticides can also alter the functioning of mitochondria, organelles closely related to maintaining calcium homeostasis. On the one hand, under conditions of oxidative stress, excess ROS causes damage and induces depolarization of the mitochondrial membrane, which impairs the ability of these organelles to maintain Ca^+2^ inside and can cause its release to the cytosol [243]. On the other hand, the sustained opening of certain nAChRs and the excessive accumulation of Ca^+2^ can cause an increase in its levels in the mitochondria, which, in turn, can cause mitochondrial inflammation and the release of apoptotic factors to the cytosol [245].

In this way, the pathological conditions induced by exposure to NN compounds, such as dysregulation of Ca^+2^ homeostasis, oxidative stress, inflammation or mitochondrial dysfunction, could eventually lead to the death of neurons in the CNS (Figure 3). However, it appears that not all brain regions are equally vulnerable to NN-induced damage. Previous studies have shown that different types of neurons exhibit different sensitivities to Ca^+2^ dysregulation, with the hippocampal and cortical neurons presenting the greatest vulnerability [243]. These findings are consistent with the results analyzed here, since several studies have shown that exposure to NNs produced important structural and functional alterations in the hippocampus. These results suggest that the hippocampus could be especially vulnerable to exposure to NN pesticides, which is likely related to the number and types of nAChRs found in this region.

The broad range of effects induced by exposure to NN pesticides, such as the alteration of neurotransmission systems, can ultimately lead to the manifestation of behavioral and cognitive changes. Thus, the results described suggest that this class of insecticides could cause significant alterations in sensorimotor functions, mood, anxiety or social behavior, as well as cause deficits in orientation, learning and memory processes.

On the other hand, when exposure takes place during the prenatal or early postnatal period, the NNs can deregulate the expression of genes essential for proper neuronal development, resulting in the manifestation of alterations in the number or morphology of these cells. These disturbances derived from early exposure could favor the appearance of developmental disorders of the nervous system or deficits in intellectual capacity.

The results obtained in bats are also of special relevance, since it has been shown that exposure to NNs can seriously affect their echolocation system, which plays a fundamental role in navigation and search for food. Thus, environmental contamination with NN pesticides could have serious repercussions for the survival of these mammals.

The findings collected here show that the effects produced by NN pesticides are not limited to their action on insect nAChRs but are also derived from their interaction with mammalian nAChRs, generating important neurotoxic effects on the nervous system of the different species studied, including humans. However, there are few published studies to date, showing the need for more research to clarify the biochemical and molecular mechanisms by which these pesticides alter the function and structure of the nervous system. This is especially relevant in cases of prolonged exposures to low concentrations of NNs, which are the conditions to which a large part of the general population is exposed, and which can pose a serious risk to their health.

#### Relevance of Daily Human Exposure to Pesticides NNs

To finalize this review, it is necessary to refer to the possible implications that exposure to the actual amounts of NNs present in the environment may have on human health. In recent years, environmental concentrations of NNs ranging from 0.001–320 μg/L have been detected in surface runoff from soils, streams and groundwater [246,247]. In addition, various studies have demonstrated the persistence of these pesticides in agricultural soils, where they can reach half-lives that vary between 9–1250 days for IMI, 6–3001 days for TMX and 17–6931 days for CLO [248]. Therefore, the increasing use of products containing NNs and their repeated application to crops has been linked to increasing levels of residues in the soil, which are indicative of environmental accumulation of pesticides throughout of the years [247].

In addition, the presence of NNs and their metabolites has been detected in numerous varieties of fruits and vegetables commonly consumed, as well as in drinking water and bovine milk [17,249]. Therefore, the general population is continually exposed to these pesticides. This has been confirmed in several studies in which the presence of these compounds and their metabolites was detected in the hair and urine of the general population [250]. While the concentrations of NNs to which humans are exposed have not been fully established, some studies have reported concentrations in human urine ranging from 0.03–2.27 μg/L of NNs and/or their metabolites [42,249].

However, it has been observed that the mean concentrations of NNs in urine have maintained a constant growth during the last years, reflecting an increasingly widespread use [249]. This can pose a serious risk, since the bioaccumulation of these compounds could occur in humans because of the consecutive ingestion of food and water contaminated with NNs. Although information on the bioaccumulation of NNs in humans is extremely limited, several animal studies have shown the ability of these pesticides to accumulate in a dose-dependent manner in the body after chronic exposure [60,251].

It appears that current environmental concentrations of NNs are relatively low and may not pose a health risk to adult humans. However, they could pose a risk to fetuses, as human and rodent studies have shown that both NNs and their metabolites can be transferred rapidly through the placenta and accumulate in the fetus [57,250,252]. Since during brain development the BBB of fetuses and neonates is immature, this will facilitate the penetration and accumulation of different chemicals, such as pesticides, in their brain [253].

On the other hand, it has been shown that some of the metabolites of NNs could persist for a longer time in the brain of mammals, mainly in the elderly. These compounds accumulate in lipid-rich tissue, such as brain tissue, and some of their metabolites have a higher affinity for mammalian nAChRs compared to the parent compound [254]. Therefore, the chronic exposure to which most humans are subjected could lead to the accumulation of these pesticides in the brain and increase the risk of neurological disorders. This could especially affect the elderly, due to their greater difficulty in metabolizing and eliminating these compounds. Thus, children and the elderly appear to be particularly susceptible to the toxicity of NNs pesticides, so the use of these compounds could be a public concern.

An important aspect that should be reconsidered is the current configuration of acceptable levels of NNs, which vary greatly from one country to another. For example, while in the European Union the maximum limit for IMI residues in tea is 0.05 mg/kg, in China this limit is ten times higher, specifically, 0.5 mg/kg [249]. Considering these values, it is evident that most of the research analyzed in this review used higher doses to study the mechanisms of toxicity of pesticides. However, in the study by Burke et al. [57] a dose of 0.5 mg/kg of IMI was administered to mice from pregnancy and pesticide was detected in the brains of mothers and offspring. This dose of pesticide was related to the appearance of alterations in the behavior and mood of the offspring. Similarly, the same dose of IMI also produced behavioral alterations in the study by Khalil et al. [71]. These results show that the consumption of NNs in amounts that are within the legal limits allowed in some countries could pose a significant risk to health. Therefore, future studies should attempt to determine the lowest dose of each NN pesticide subtype that is capable of inducing biochemical and behavioral abnormalities, as well as concomitantly quantifying tissue levels and their excretion in urine.

In short, NNs were traditionally considered to have less toxicity in mammals due to their poor penetration through the BBB and their lower affinity for mammalian nAChRs. This assumption has been widely accepted and has led to an increasing use of this class of pesticides in modern agriculture. However, the results analyzed in the present review contradict this notion and demonstrate that these compounds can cross the BBB and that some of their metabolites could have a higher affinity for mammalian nAChRs than the parent compound. Consequently, NNs pesticide residues could accumulate in the human brain and cause important neurological disorders, especially in infants and the elderly. For this reason, these findings suggest the need to introduce modifications in the accepted limits for NNs in some countries and adopt more conservative values. Above all, taking into account that fetuses are also exposed to pesticides and their metabolites at levels similar to those of the mother.

## 4. Limitations

The main limitation of the present systematic review is that most of the findings on the neurotoxic effects of NN pesticides in mammals have been obtained from studies in rodents. Therefore, various methodological aspects must be considered before extrapolating the results obtained using animal models to human problems. A central issue with generalizing results is found in the differences between species, doses of pesticides, exposure times, and in the routes of administration used. Regarding the routes of administration, studies have focused mainly on investigating the effects of oral administration of pesticides, a route of exposure that can be very useful to assess the effects of consuming food or water contaminated with NNs. However, other routes of exposure to NN pesticides have not been sufficiently investigated, such as the dermal route or the respiratory route.

On the other hand, although there are central elements and developmental patterns that are common for the different mammalian species, a clear equivalence between the periods of development of the rodent brain with the same periods in the human brain development has not been demonstrated [255]. For example, studies in which exposure to NN pesticides occurs during the early postnatal period of rodents, would correspond to the third trimester of gestation in humans [242]. Therefore, these considerations must be taken into account when generalizing the results described.

Finally, many of the studies included here show the effects derived from exposure to doses of NNs greater than those found in the environment or that the general population is routinely exposed. Therefore, exposure to lower doses of NN compounds may not cause the wide range of neurotoxic effects documented here, or at least not produce them with the same severity.

## 5. Conclusions

The studies analyzed in this review indicate that exposure to NN pesticides can pose a risk to the integrity and functioning of the nervous system of different species of mammals, including humans. This neurotoxicity seems to originate from their action on nAChRs, especially the α4β2 and α7 subtypes, which are the most abundant in the CNS. Prolonged exposure to NNs can cause the sustained opening of some nAChRs, especially α7 subtype, which maintain their activity in the sustained presence of agonists and are highly permeable to Ca^+2^. Increased levels of Ca^+2^ can activate or inhibit a number of intracellular signaling pathways in both neurons and glial cells, leading to alterations in neurotransmission, oxidative stress or inflammation, which can further enhance neurotoxic conditions inside the cell. The activation of different apoptotic pathways ultimately leads to neuronal death.

The neurotoxic effects of NN pesticides appear to affect different brain regions differently, with the hippocampus being the most vulnerable. Thus, the damage caused by NN compounds is manifested in mammals with alterations in motor behavior, mood, anxiety, social behavior, as well as with serious deficiencies in the cognitive processes of orientation, learning and memory. These alterations can pose a clear risk to the survival of mammals and to human health, which is why future research needs to focus on knowing their true scope.

## Figures and Tables

**Figure 1 ijms-22-08413-f001:**
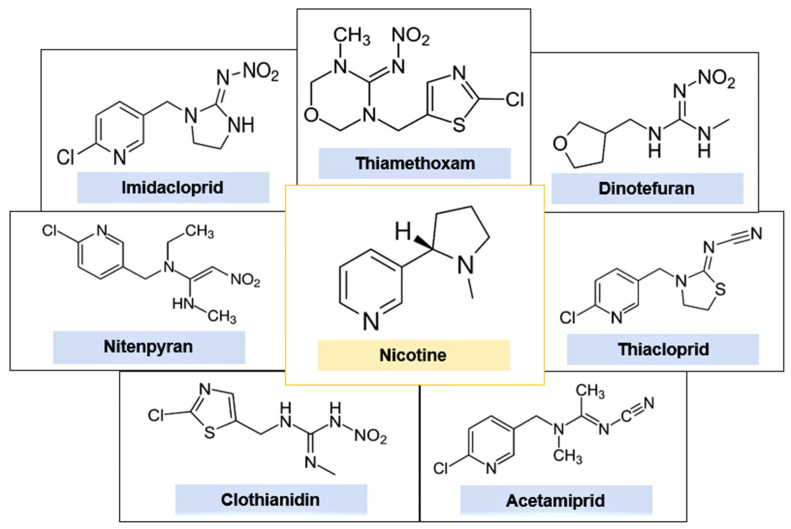
Chemical structures of nicotine and the neonicotinoid insecticides.

**Figure 2 ijms-22-08413-f002:**
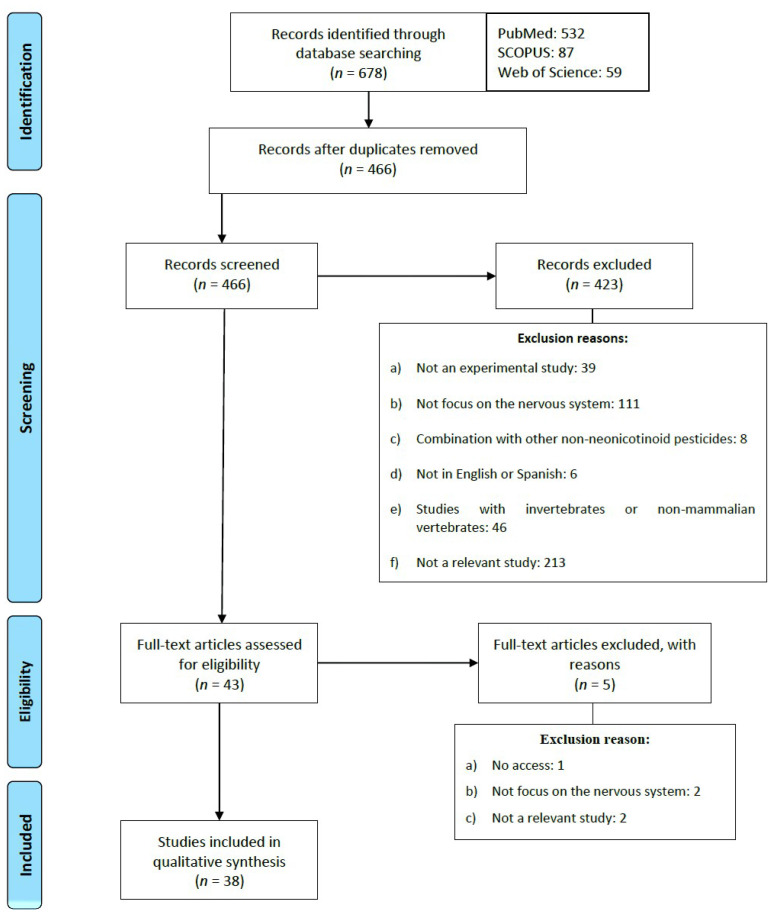
Flow diagram of the systematic search process.

**Figure 3 ijms-22-08413-f003:**
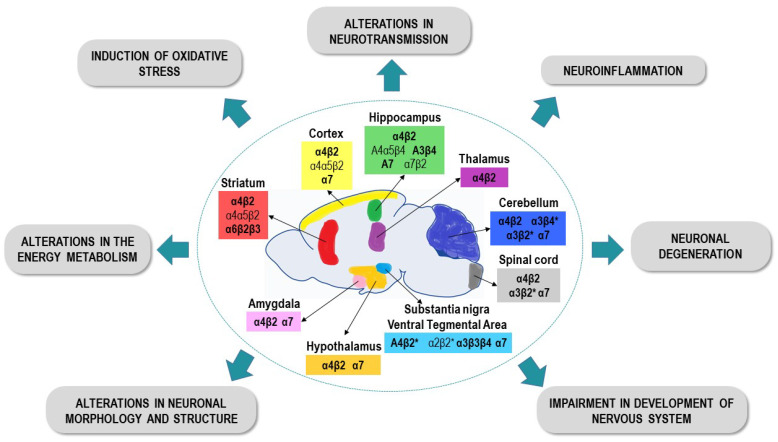
Distribution of the nAChR subtypes identified in some regions of the rodent CNS, and the main toxic effects observed after exposure to NNs pesticides in the rodent CNS.

**Figure 4 ijms-22-08413-f004:**
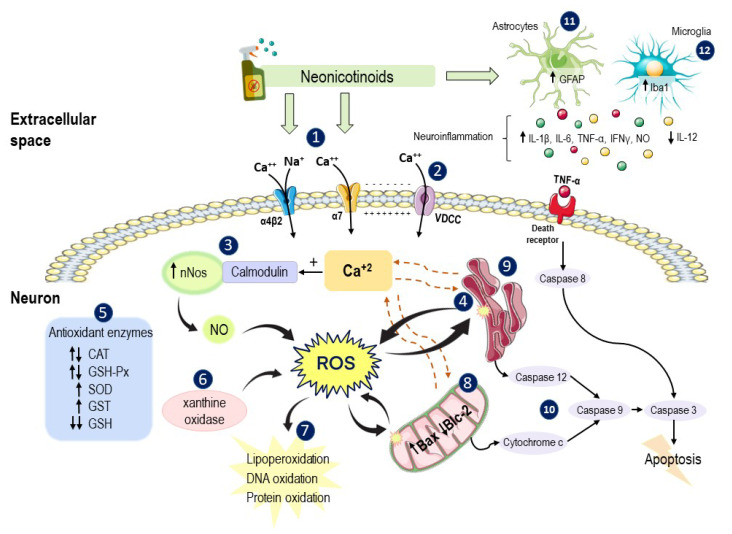
Main mechanisms of neurotoxicity proposed for NN pesticides in the nervous system of rodents, human, and other mammals. NNs induce activation of α4β2 and α7 subtype receptors (1), increased Ca^+2^ influx, membrane depolarization, and VDCC opening (2), with a greater increase in Ca^+2^ levels. Ca^+2^ can bind to calmodulin, increasing NO synthesis (3) and ROS production, and it can also bind to ER ryanodine receptors (4), further increasing its concentrations in the cytosol. NNs also modify the activity of several antioxidant enzymes (5) and increase the activity of xanthine oxidase (6) and, consequently, there is an increase in ROS production. Increased levels of ROS lead to LPO, DNA and protein oxidation (7), and severely affect mitochondria (8) and ER (9), activating two different pathways of cellular apoptosis (10). NNs also induce astrocyte and microglial activation, with upregulation of GFAP (11) and Iba1 (12), respectively, and increased release of pro-inflammatory cytokines and ROS. Some of these molecules, such as TNF-α, can bind to death receptors and trigger neuronal apoptosis (13). Parts of the figure were created using templates from Servier Medical Art, which are licensed under a Creative Commons Attribution 3.0 Unported License (http://smart.servier.com/ accessed on: 6 July 2021). Abbreviations: VDCC: voltage-dependent calcium channels; nNOS: neuronal nitric oxide synthase; NO: nitric oxide; ROS: reactive oxygen species; CAT: catalase; GSH-Px: glutathione peroxidase; SOD: superoxide dismutase; GST: Glutathione S-transferase; GSH: glutathione; GFAP: glial fibrillary acidic protein; IL-12: interleukin 12; IL-1β: interleukin 1beta; IL-6: interleukin 6; TNF-α: tumor necrosis factor alpha.

**Table 1 ijms-22-08413-t001:** Neurotoxic effects derived from in vivo exposure to neonicotinoid pesticides in rodents.

Species	Dose and Time Exposition	Objectives	Results	Reference
Wistar rats	IMI: 0.5, 2 or 8 mg/kg/day orally for 3 months	Evaluate the effects of different doses of IMI on learning and memory in infant and adult rats	- Treatment with 2 and 8 mg/kg induced a decrease in learning in infant rats and with the 8 mg/kg dose in adult rats - Increased M1 receptor expression at doses of 2 and 8 mg/kg in adult rats.	[11]
Wistar rats	IMI: 0.06, 0.8 or 2.25 mg/kg/day orally for 28 days	Investigate the effects of IMI on cholinesterase activities, oxidative stress biomarkers and primary DNA damage in blood and brain tissue	- There were no alterations in total cholinesterase, AChE or butyrylcholinesterase activity in the brain - There was no change in CAT and SOD activity, but there was an increase in GSH-Px activity - Presence of IMI in brain tissue of animals treated with the two highest doses - Dose-dependent neuronal DNA damage	[15]
Wistar rats	IMI: 20 mg/kg orally Single dose	Evaluate pharmacokinetic and pharmacodynamic responses after single oral exposure	- The maximum concentration of IMI and its metabolites in the brain was observed at 12 h after exposure - Inhibition of brain AChE at 6–48 h after administration	[59]
Wistar rats	IMI: 10 or 20 mg/kg/day orally for 60 days	Assess the alterations induced by IMI in the biochemical, histopathological and protein profile in plasma and brain	- Decreased brain AChE activity	[117]
Wistar rats	IMI: 1 mg/kg/day orally for 30 days	Assess the effects of chronic exposure to IMI on the induction of oxidative stress and inflammation	- Increased NO production - Improved transcription of iNOS, eNOS and nNOS mRNA - Increase in LPO - Increased activity of xanthine oxidase - Increased CAT activity and decreased GSH levels - Increased expression of TNF-α, IL-1β, IL-6, and IFN-γ, but decreased IL-12	[128]
Sprague-Dawley rats	IMI: 1 mg/kg/day orally for 60 days	Evaluate the effects of IMI on neurobehavioral performance, oxidative stress and the induction of apoptosis in the brain of adult or adolescent rats	- Less exploratory activity, deficit of sensorimotor functions and depression - Reduction of serotonin, GABA and dopamine - Increase in the levels of protein carbonyl, 8-hidroxyguanosine and MDA, but reduction in the total antioxidant capacity - Neural degeneration - Increased expression of GFAP - Increase in Bax levels and decrease in Bcl-2 levels	[70]
Sprague-Dawley rats	IMI: 0.5 or 1 mg/kg/day orally for 60 days	Study the effects of IMI on stress by assessing cortisone and catecholamine levels, with a focus on behavioral alterations	- Behavioral deficits, particularly at the highest dose of IMI	[71]
CD-1 mice	IMI: 0.5 mg/kg/day infusion through an osmotic pump From GD4 until PND21	Assess the effects of IMI after an intrauterine and early postnatal exposure	- The offspring of mothers treated with IMI showed elevated motor activity, improved social dominance, reduced depressive behavior and a decrease in social aggression - Low levels of IMI were detected in the brains of treated mothers and traces of the pesticide in the brains of some pups after exposure to weaning	[57]
KM mice	IMI: 5 or 20 mg/kg/day orally for 28 days	Examine the histopathological, biochemical and metabolic alterations induced by IMI in the hippocampus and liver	- Hippocampal damage in animals treated with the highest dose - Alteration in the metabolic profile in the hippocampus at both doses	[112]
Wistar rats	ACE: 10, 20 or 40 mg/kg/day orally for 28 days	Investigate the effect of ACE on spatial memory and the vulnerability of the hippocampal glutamatergic system	- Alteration of learning, consolidation and memory retrieval processes, especially at the highest doses - Dose-dependent increase of pesticide concentration in the hippocampus - Neuronal degeneration and apoptosis in the dentate gyrus - Reduced glutamate levels - Reduced expression of NR1, NR2A and NR2B genes after exposure to the intermediate dose of ACE	[60]
Wistar rats	ACE: 3.14 mg/kg/day orally for 6 months	Assess the effects of ACE on membrane integrity and mitochondrial potential	- Decreased GSH levels and GSH-Px and CAT activities in mitochondria - Increased GST and SOD activity in the mitochondrial matrix - Increased levels of MDA - Increased membrane permeability and mitochondrial swelling and significant decrease in mitochondrial respiration (O_2_ consumption)	[156]
A/J mice	ACE: 71 or 710 μg/g/day orally for 3 and 7 days	Investigate the accumulation of ACE and expression of nAChRs in different areas of the brain	- Treatment with the pesticide produced no effects on the histology or on the expression of CD34 - Presence of higher concentrations of ACE in the midbrain - Decreased expression of nAChRs, especially those of the β2 subtype, in various brain regions of animals treated for 7 days with the highest dose of ACE	[61]
ICR mice	ACE: 5 mg/kg/day orally. From GD6 until GD18	Evaluate the effects of repeated maternal exposure to ACE on the neurodevelopment of the offspring	- Cortical hypoplasia, decreased neurogenesis and abnormal neuronal distribution in the neocortex - Increased number of microglial cells immunoreactive for Iba1 and amoeboid-like microglial cells - Increased activation of the M1 microglial phenotype	[65]
ICR mice	ACE, IMI: 5 mg/kg/day orally From PND12 until PND26	Evaluate the effects of ACE and IMI exposure on neurogenesis and microglial profiles in the dentate gyrus of the developing hippocampus	- Exposure to the two pesticides reduced neurogenesis in the hippocampal dentate gyrus - Both ACE and IMI increased the number of activated amoeboid and M1-type microglial cells.	[66]
Sprague-Dawley rats	CLO: 3.5 mM by local administration through a microdialysis probe	Determine the neurochemical effects and mechanisms of action of CLO on striatal dopamine release	- Increase in the in vivo release of striatal dopamine in a concentration-dependent manner - The increases were dependent on the presence of Ca^+2^ in the extracellular medium, the depolarization of the membrane, the vesicular storage of dopamine and independent of the dopamine transporter	[134]
Sprague-Dawley rats	CLO: 150 or 300 μmol by local administration through a microdialysis probe	Evaluate the role of some subtypes of nAChRs and mAChRs in CLO-induced striatal dopamine release	- Administration of selective α4β2 or α7-receptor antagonists decreased CLO-induced dopamine release in vivo - Pretreatment with a β2-subunit antagonist did not affect dopamine release - Pretreatment with an antagonist of mAChR blocked CLO-induced increases in extracellular dopamine levels	[135]
CD-1 mice	CLO: 0.003%, 0.006% or 0.012% orally. From 5 weeks of age of the F0 generation to 11 weeks of age of the F1 generation	Assess the effects of CLO exposure on reproduction and behavior over different generations	- Appearance of various adverse effects on neurobehavioral parameters - Adverse effects on exploratory behavior in mice of generations F0 and F1 - Although the pesticide produced adverse effects in males and females, these effects differed according to sex	[68]
CD-1 mice	CLO: 0.002%, 0.006% or 0.018% orally. Gestation and lactation periods	Assess the neurobehavioral effects of maternal exposure to CLO	- Occurrence of several adverse effects on offspring behavior - Adverse effects on exploratory and spontaneous behavior of the offspring with the intermediate dose of CLO - Although the pesticide produced adverse effects in males and females, these effects differed according to sex	[69]
C57BL/6J mice	CLO: 5 mg/kg orally Single dose	Investigate the role of aging in CLO-induced behavioral effects	- Decreased locomotor activity in aged mice, but not in adult mice - Higher concentrations of CLO and its metabolites in the brains of aged mice than in adult mice	[58]
C57BL/6N mice	CLO: 5 or 50 mg/kg orally Single dose	Evaluate the neurobehavioral effects of CLO and explore the brain regions targeted by neonicotinoids in mammals	- Induction of audible vocalizations (4–16 kHz) for highest dose - Anxiety-like behavior	[78]
C57BL/6N mice	DIN: 100, 500 or 2500 mg/kg/day orally for 6 weeks	Analyze the biochemical and behavioral effects of DIN exposure during the peripubertal period on the nigrostriatal pathway	- Increased locomotor activity in a dose-dependent manner, but no change in anxiety-like behavior - Increased TH expression in the *substantia nigra*.	[74]
C57BL/6NCrSlc mice	DIN: 100, 500 or 2500 mg/kg/day orally for 5 weeks	Investigate the relationship between subchronic DIN exposure and a depression-related phenotype	- Antidepressant activity was observed in the tail suspension test - No decrease in the number of serotonergic cells was observed	[87]

Abbreviations. IMI: imidacloprid; MDA: malondialdehyde; GFAP: glial fibrillary acidic protein; GD: gestational day; PND: postnatal day; NO: nitric oxide; iNOS: inducible nitric oxide synthase; eNOS: endothelial nitric oxide synthase; nNOS: neuronal nitric oxide synthase; LPO: lipid peroxidation; CAT: catalase; GSH: glutathione; TNF-α: tumor necrosis factor alpha; IL-1β: interleukin 1beta; IL-6: interleukin-6; IFN-γ: interferon gamma; IL-12: interleukin-12; CLO: clothianidin; nAChR: nicotinic acetylcholine receptor; mAChR: muscarinic acetylcholine receptor; ACE: acetamiprid; GSH-Px: glutathione peroxidase; GST: glutathione S-transferase; SOD: superoxide dismutase; Iba1: ionized calcium-binding adapter molecule 1; AChE: acetylcholinesterase; DIN: dinotefuran; CD34: cluster of differentiation 34; TH: tyrosine hydroxylase.

**Table 2 ijms-22-08413-t002:** Neurotoxic effects observed after in vitro exposure to neonicotinoid pesticides in rodents.

Cellular Line	Dose and Time of Exposure	Objective	Results	Reference
Neuron-enriched cultures from neonatal rat cerebellum	ACE, IMI: 1 μM for 14 days	Examine the neurotoxic effects of long-term and low-dose exposure on cultures enriched with cerebellar neurons	- There were no alterations in the number or morphology of immature neurons or glial cells - Altered dendritic arborization of Purkinje cells after treatment with NNs - IMI and ACE altered expression of genes essential for neurodevelopment	[64]
Primary cultures of cerebellar neurons from neonatal rats	ACE, IMI: 1, 10 or 100 µM	Determine the effects of the two NNs on neuronal nAChRs and compare their effects with nicotine	- All ACE and IMI concentrations used caused increases in Ca^+2^ influx into cerebellar neurons (mainly granule cells)	[120]
PC12 cells	IMI, ACE, CLO, TMX: 1–100 μM for 5 days	Evaluate the neurotoxic effects on the development of several commonly used pesticides	- IMI showed a dose-dependent inhibitory effect on neurite outgrowth. - ACE (10 µM) produced significant alterations in the transcription of genes related to the development of the nervous system.	[63]
PC12 cells	IMI: 1, 3, 30 or 100 μM for 24 and 48 h	Assess the effects of IMI on the catecholaminergic function of chromaffin cells	- IMI improved the synthesis of catecholamines by increasing the expression of the genes that synthesize TH and PNMT - IMI facilitated this transcription by acting as a partial agonist of the α3β4 and α7 nAChR subtypes that, when activated, induced a long-term activation of the intracellular signaling pathways PKA/CREB and RhoA	[136]
PC12 cells	ACE: 100–700 μM for 24 h	Investigate the toxic effects of ACE on PC12 cells	- Increased cell death by apoptosis - Increased ROS and LPO levels - Reduction in mitochondrial membrane potential - Dose-dependent increase in DNA damage	[155]
NSPC y N2a	IMI, CLO: 0–4000 μΜ for 48 h	Investigate the toxic effects and metabolic changes induced by IMI, CLO and their mixture in cell cultures	- Alteration in metabolism and energy balance of NSPCs with both insecticides alone or mixed	[189]

Abbreviations. ACE: acetamiprid; ROS: reactive oxygen species; LPO: lipid peroxidation; NN: neonicotinoid; IMI: imidacloprid; TMX: thiametoxam; nAChR: nicotinic acetylcholine receptor; CLO: clothianidin; NSPC: neural stem/progenitor cells; N2a: mouse neuroblastoma-derived Neuro-2a cells; TH: tyrosine hydroxylase; PNMT: phenylethanolamine N-methyltransferase; PKA: protein kinase A; CREB: cAMP response element-binding; RhoA: ras homolog family member A.

**Table 3 ijms-22-08413-t003:** Effects of exposure to neonicotinoid pesticides in humans.

Type of Study	Toxic Agent	Exposure Mode	Results	Reference
Case study	THI	Oral (attempted suicide)	- Status epilepticus (multiple episodes of generalized tonic-clonic seizures) - State of deep unconsciousness	[208]
Case-control study	ACE, IMI, CLO, NIT, THI, and TMX	Consumption of food with pesticides	- A relationship was found between the occurrence of nicotinic symptoms in the general population and environmental exposure to ACE and TMX.	[209]
Cohort study	IMI	Residential exposure	- Significant inverse associations were found between large-scale children’s IQ and close agricultural use of IMI	[213]
Case-control study	IMI	Use of IMI on pets	- The odds of reported prenatal exposure to IMI among mothers of children with ASD were twice that of mothers of typically developing children.	[214]

Abbreviations. IMI: imidacloprid; NIT: nitenpyran; THI: thiacloprid; TMX: thiametoxam; ASD: autism spectrum disorders; ACE: acetamiprid; CLO: clothianidin.

**Table 4 ijms-22-08413-t004:** Effects of neonicotinoid pesticides in human cells observed in vitro.

Cellular Line	Dose and Time of Exposure	Objective	Results	Reference
SH-SY5Y	CLO: 1–100 μM for 24 h	Assess whether CLO could affect the structure or function of the human nervous system	- CLO increased the number of cells and the growth of their neurites through its action on nAChRs - Induced the entry of Ca^+2^ to the intracellular medium and the phosphorylation of ERK - Negatively regulated genes related to neuronal function and morphology	[200]
SK-N-SH	IMI, ACE, TMX: 10–2000 mg/L for 24 h	Study the toxicity of NNs alone or in combination	- NNs alone or in combination inhibited growth and reduced cell viability, with TMX being the most toxic	[204]
SH-SY5Y	IMI, ACE, CLO, THI, TMX: 0.05–4 mM for 24 and 48 h	Investigate the possible effects of common NN insecticides on cytotoxicity and DNA damage	- Concentrations greater than 100 μM of THI and TMX produced DNA damage, with THI being the most cytotoxic insecticide - The five NNs produced DNA damage at concentrations of 500 μM	[205]

Abbreviations. IMI: imidacloprid; ACE: acetamiprid; TMX: thiametoxam; NN: neonicotinoid; CLO: clothianidin; nAChR: nicotinic acetylcholine receptor; THI: thiacloprid; TMX: thiametoxam; ERK: extracellular signal-regulated kinase.

**Table 5 ijms-22-08413-t005:** Neurotoxic effects of neonicotinoid pesticides in other mammalian species.

Species	Dose and Time Exposition	Objectives	Results	Reference
Deer *(Odocoileus virginianus)*	IMI: 1.500, 3.000 or 15.000 ng/L orally From May to October	Assess the toxic effects of IMI in adult female white-tailed deer	- IMI was present in the organs of the control group animals - IMI values in the brain were low or undocumented	[219]
Bat *(Hipposiderosarmiger terasensis)*	IMI: 20 mg/kg orally for 5 days	Compare spatial memory of bats before and after chronic treatment with a low dose of IMI	- The stereotypical flight patterns of the treated bats were different from their originally learned paths - Increased Bax/Bcl-2 ratio and levels of caspase 3 and caspase 3 cleaved in the CA1 hippocampus and areas of the medial entorhinal cortex	[111]
Bat *(Hipposideros armiger terasensis)*	IMI: 0.5 mg orally for 5 days	Examine whether IMI toxicity can interfere with the echolocation system of bats	- The bats were disoriented after treatment - Decreased expression of FOXP2, prestin and otoferlin proteins - Increased expression of TNF-α - Increased expression of cytochrome C and caspases 3 and 12	[220]

Abbreviations. IMI: imidacloprid; FOXP2: Forkhead box protein P2; TNF-α: tumor necrosis factor alpha.

## Data Availability

Not applicable.

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
