# Peer review of "Neurotoxic Effects of Neonicotinoids on Mammals: What Is There beyond the Activation of Nicotinic Acetylcholine Receptors?—A Systematic Review"

_ijms, 2021, doi:10.3390/ijms22168413_

Round 1

Reviewer 1 Report

Reviewing manuscript n° 1317758

 “Neurotoxic effects of neonicotinoids on mammals: what is there beyond the activation of nicotinic acetylcholine receptors?” by Costas-Ferreira & Faro

The manuscript focuses on the impact of neonicotinoid (NN) insecticides on mammalian nAChRs receptors. The review shows through original scientific articles (38 articles selected) that NN  compounds have: (i) potential toxic effects on the nervous system development of mammals; (ii) some consequences on behavior and cognitive function; (iii) impairments on neurotransmission processes especially Ca2+ homeostasis and (iv) consequences on  inflammation.  

The authors have listed the effects of NN both in experimental studies conducted in rodent models and cell cultures, in other mamals, as well as human effects through epidemiological studies. The manuscript is clearly written and easy to read. The manuscript is also well structured and paragraphs are well described. The topic is of great importance regarding the potential impact of NN on human health. It contains 5 tables and 3 figures.  

I consider that this manuscript is suitable for publication in the IJMS journal after minor revisions. 

Here are my suggestions:  

Table 1: The references listed are all mixed, you have alternatively, rats, then mice, and rats again.

The references in Table 1 should be reorganized as following:

 First by specific NN : IMI, ACE, CLO, DIN.

Second: for each drug, reorganize by species: rats (Winstar and Sprague), WT mice and Tg mice

Table 1 and Table 5: Specie should be written “species” otherwise the word in singular means money

Table 2: The same comment as above, you should reorganize the references.First: primary culture, then cell lines (PC12 and N2a)

Table 5: You should reorganize the references by species: first Odocoileus v. and then Hipposideros a. 

Please revised the form of some sentences in the text to be more straightforward: for instance: page 10 line 606 “ In the present review, only three recent studies have been identified that study the effect of….” Could be “Only three studies have described the role of NN on mammals other than rodent and humans” 

another example in the Discussion section: page 12, line 693- page 13, line 718: 712-714: this paragraph contains some repetitions, maybe suppress the sentence “ as previoulsly mentioned, these receptors are highly permeable to Ca2+, ….large amount of this ion into the cell” to make the paragraph more straightforward. 

Figure 3: the size is too small, this figure should be enlarged.

Reviewer 2 Report

This is a sound review about the adverse effects of neonicotinoid pesticides on non-target species. The review not only covers a description of the adverse effects but also contains a very in deep analysis of molecular and cellular mechanisms of toxicity. The information is presented in a very comprehensible way and, on the contrary to other long reviews, the reading of this manuscript is not tedious.

In conclusions, I consider the manuscript of high potential interest to the readers. I have no major objections but I would like to address to the way in which the information is presented rather than to the content as itself.

I wonder whether the authors have scored in some way the quality of the assessed papers in order to differentiate those providing stronger evidences form those providing weaker evidences. Some examples of criteria that could help to score, whether dose-response is assessed or not, robustness of the statistical assessment, independent replicates in in vitro experiments, etc.

In first sentence of discussion: “…exposure to NN pesticides induces important toxic effects on the nervous system…”. This sentence should be rewritten. The manuscript deals with hazard identification rather than risk assessment. Whether the toxic effect is going to appear or not is going to be dependent on the severity of the exposure.

Figure 2 is too small. Even with zoon at 150%, I cannot read the different parts of the brain inside the circle.

I suggest the following changes in Figure 3. First, to state that the main part of the figure corresponds to a neuron. Second, to introduce a change showing that neonicotinoids also interact with astrocyte and microglia provoking releasing of neuroinflammation mediators. In the current design, it seems that neuron insult provokes release of cytokine and others instead of via astrocyte and microglia activation.

Just for facilitating the reading, please, introduce in Table 5 in the column entitled species the entry bat or deer just before the scientific name.

Minor suggestion. I think that it could be illustrating whether the authors introduce an additional Figure showing the chemical structure of nicotine and the chemical structure of the main pesticides belonging to neonicotinoid family.

Reviewer 3 Report

The manuscript ijms-1317758, entitled “Neurotoxic effects of neonicotinoids on mammals: what is there beyond the activation of nicotinic acetylcholine receptors?” is a well written and very comprehensive review of the literature on neonicotinoid pesticides and their toxicity in mammals.  Overall, I would only have minor suggestions, but one more significant question is:

The authors state that ‘large quantities’ of NN compounds are needed.  Assuming this is due to the lack of crossing the blood-brain barrier.  In general, NN compounds have elicited more peripheral side effects.  There are spots early in the review when these topics were mentioned, but not expanded upon. I would suggest:

1. Have a section where sample NN compounds are listed and their ability to cross the BBB is assessed (whether it is a fractional expression of the total dose administered or through in vitro studies that estimate the ability of different NN compounds to cross the BBB

2. Since the inference is that an individual would have to be exposed to very large concentrations – what is the relevance to what would happen ‘real-world’? Also, what is the ability of these NN compounds listed to bioaccumulate either in the environment or in a mammal?

These additions would add to the overall information that is presented in this review.  Additionally, I do have some minor housekeeping issues (listed with the corresponding line #).  The majority of the suggestions just involved shortening sentences and streamlining some of the text to improve clarity and readability.

42 – delete the 2nd ‘in’ before pets (redundant)

80 – delete ‘In fact’ – starting the sentence with Epidemiological…

97-98 – the sentence containing “other mammals, although in recent years other possible mechanisms by which these pesticides produce their toxic effects on the nervous system have been investigated.” Could be rewritten for clarity to “other mammals, although other possible mechanisms by which these pesticides produce their toxic effects on the nervous system have been investigated in recent years.”

Table 1 after line 140 – Species and Objectives are misspelled

The objectives column could be redone and shortened.  Some are wordy for a little space.  An example is for A/J mice. The Objective “Investigate the accumulation of ACE in different areas of the brain and to evaluate its effects on the expression of nAChRs in these regions” which could be shortened to “Investigate the accumulation of ACE and expression of nAChRs in different areas of the brain”

Table 1 ref 135 line – change ‘had no effect on’ dopamine release to ‘did not affect’ dopamine release

156 – “Some of the studies analyzed here confirm that NNs cross the BBB, since the presence of these pesticides and their metabolites” is wordy – reduce to “The studies analyzed confirmed that NNs do cross the BBB, based on the detection of pesticides and their metabolites in the brains of exposed animals…” for clarity.

206 – maybe change ‘stress’ to ‘stressful’.  There seems to be something missing when it is just written as ‘stress’

234 – change “has been shown to be” to ‘is’

258 – change “is involved in modulating’ to ‘modulates’

329 – change ‘brain of adult…’ to ‘brains from adult…’

349 – change “have been shown to be” to ‘are’

381 – change “which can lead’ to ‘leading’

533 – delete ‘that are’

535 – delete ‘included here’

540 – change ‘as a result of’ to ‘due to’

558 – delete ‘in’ before ‘the growth’

565 – change ‘are able to’ to ‘can’

Table 5 – between lines 612 and 613 has some typos.  Species is missing an ‘s’, and ‘Objectives’ is missing a ‘c’.

629 – delete ‘in’ before ‘the medial’

708 – change “it is possible that this prolonged exposure induces” to ‘prolonged exposure may induce’

797-800 – “The main problem when generalizing the results is found in the biological differences between the species, as well as the differences in the doses of pesticides, in the exposure times, and in the administration routes used.” Seems to be a little awkward.  Consider shortening to:

‘A central issue with generalizing results is found in the differences between species, doses of pesticides, exposure times, and in the routes of administration used.’

803-804 – shorten “However, there are other routes of exposure to NN pesticides that have not been sufficiently investigated, such as the dermal route or the respiratory route.”

To: “However, other routes of exposure to NN pesticides have not been sufficiently investigated, such as the dermal route or the respiratory route.”
